# Zirconia Dental Implants Surface Electric Stimulation Impact on *Staphylococcus aureus*

**DOI:** 10.3390/ijms25115719

**Published:** 2024-05-24

**Authors:** Flávio Rodrigues, Helena F. Pereira, João Pinto, Jorge Padrão, Andrea Zille, Filipe S. Silva, Óscar Carvalho, Sara Madeira

**Affiliations:** 1Center for MicroElectroMechanical Systems (CMEMS), University of Minho, 4800-058 Guimarães, Portugal; gabriel_rodrigues97@hotmail.com (F.R.); helena.frs.pereira@gmail.com (H.F.P.); id8102@alunos.uminho.pt (J.P.); fsamuel@dem.uminho.pt (F.S.S.); oscar.carvalho@dem.uminho.pt (Ó.C.); saramadeira@dem.uminho.pt (S.M.); 2Associate Laboratory in Biotechnology and Bioengineering and Microelectromechanical Systems (LABBELS), 4800-058 Guimarães, Portugal; 3MIT Portugal Program, School of Engineering, University of Minho, Campus de Azurém, 4800-058 Guimarães, Portugal; 4Center for Textile Science and Technology (2C2T), University of Minho, 4800-058 Guimarães, Portugal; azille@2c2t.uminho.pt

**Keywords:** dental implant, biofilms, *Staphylococcus aureus*, electric current

## Abstract

Tooth loss during the lifetime of an individual is common. A strategy to treat partial or complete edentulous patients is the placement of dental implants. However, dental implants are subject to bacterial colonization and biofilm formation, which cause an infection named peri-implantitis. The existing long-term treatments for peri-implantitis are generally inefficient. Thus, an electrical circuit was produced with zirconia (Zr) samples using a hot-pressing technique to impregnate silver (Ag) through channels and holes to create a path by LASER texturing. The obtained specimens were characterized according to vitro cytotoxicity, to ensure ZrAg non-toxicity. Furthermore, samples were inoculated with *Staphylococcus aureus* using 6.5 mA of alternating current (AC). The current was delivered using a potentiostat and the influence on the bacterial concentration was assessed. Using AC, the specimens displayed no bacterial adhesion (Log 7 reduction). The in vitro results presented in this study suggest that this kind of treatment can be an alternative and promising strategy to treat and overcome bacterial adhesion around dental implants that can evolve to biofilm.

## 1. Introduction

Tooth loss is a common problem that affects people throughout their lifetime [1,2]. Tooth decay, periodontitis, trauma, genetic disorders, aging and other oral diseases have been reported as the main causes of tooth loss [3,4,5]. Dental implants represent one of the solutions to replace missing teeth. Dental implants have been widely used for more than 40 years and are increasingly accepted by patients. For this reason, this field has become one of the fastest growing fields in contemporary dentistry [6,7,8]. However, complications and failures continue to occur in dental implants, and the attention given to this issue is increasing [9]. There are several reasons for implant failure, such as fracture of the implant or abutment, biological complications, aspects related to the surgical procedure and the patient’s healing capacity and habits [10,11,12]. The biological complications of dental implants are related to the nature and the physiology of biofilms on the surface of the implant or in its adjacent components, specifically a disease known as peri-implantitis [7,13], and have been thoroughly studied recently. In fact, following the placement, dental implants are subjected to the application of loads. An inadequate load distribution could lead to vertical bone resorption and, consequently, facilitate the entrance of bacteria and biofilm formation. This is one of many factors that can lead to diseases in the peri-implant area, such as mucositis, which then evolves into peri-implantitis associated with the continued bone loss [14]. Bone loss due to an inflammatory response to titanium particles and ions is sometimes referred to as metallosis [15]. Factors such as implant design, degree of roughness, the acute rejection, poor oral hygiene and smoking are other causes for these diseases [16,17]. Peri-implantitis is one of the most frequent complications associated with dental implants, affecting around 1 to 4 implants per 10 placed and caused by plaque bacteria, which is characterized by peri-implant soft tissue inflammation accompanied by bone loss [18,19]. There are several strategies available to deal with this infection, comprising conservative (non-surgical) and surgical strategies [20,21]. The first objective of these strategies is to reduce the bacterial adhesion to the surface of the implant and remove the biofilm if it is already formed. Bacterial adhesion is related to several factors, such as the material of the implant, its geometry and dimensions and its chemical and physical surface characteristics, e.g., roughness, wettability and microstructure, among others. In general, rougher surfaces are associated with more bacterial adhesion; however, sometimes these surfaces can reduce the bacterial adhesion [22]. Surface modification strategies, such as acid etching, grinding, polishing, blasting and laser etching, have been used to prevent adhesion and the establishment of bacteria. Specific materials can also be added to the surface materials as a coating [23,24,25]. Some coatings are carried out by chemical modification, namely the incorporation of bacteria repellent, the release of noble metals and the inclusion of proteins and antibiotics [26]. Laser etching, one of the strategies mentioned above, has been widely used in recent times. Its ability to remove material quickly and its high-precision cutting at the micro level allow the creation of reproducible patterns on the most diverse surfaces without any risk of surface contamination [27,28,29]. However, despite the surface modification efforts, bacteria still adhere to the surface. Mechanical, photodynamic, laser and chemical strategies, such as local debridement and implant surface decontamination, have been used to decrease or even eliminate the biofilm already formed [30,31]. Nevertheless, these strategies are not consistently effective, thus enhanced approaches are warranted [32]. The application of electric current has emerged as a promising strategy [33]. There are already some reports on its successful use [34,35]. The literature reports two different approaches: the bioelectric and electricidal effects. The first combines the use of electric current and antimicrobials, while the second employs only the use of electric current [34,35,36]. The electricidal effect is based on mechanisms that can detach and kill bacteria leading to a decrease in biofilm, or even its complete eradication. These mechanisms are electrostatic and repulsive forces, pH modification due to electrochemical reactions, formation of reactive species and bubbles of hydrogen and oxygen at the biofilm/electrode interface [33,36,37,38]. Titanium has been considered the gold standard material for dental implants due to its mechanical properties and biocompatibility [39,40]. However, some drawbacks have been described, such as the release of metallic ions, plaque accumulation and its dark grayish color, which although a characteristic of titanium and not a problem, can become visible through soft tissues, especially in the case of thin gingiva biotypes, affecting the implant aesthetics [41]. Zirconia (Zr) based materials are biocompatible ceramics that have been introduced in dental implantology as a promising alternative to titanium due to their properties [42]. They are chemically inert, avoiding ion release [26], have an aesthetic property, namely their color, that mimics the appearance of natural teeth [7] and, in addition, retain less plaque [43]. However, despite the advantages over titanium, these materials are non-conductive [42]. The present study aims to understand the influence of alternating electric current on *Staphylococcus aureus* adhered to the zirconia surfaces. In addition, the study assesses the cytotoxicity of Zr containing silver (Ag) to validate their biocompatibility.

## 2. Results and Discussion

### 2.1. Specimen Characterization

Channels and holes produced through LASER texturing of Zr specimens are depicted in Figure 1, which shows SEM micrographs of the top surface of laser textured channels, holes and the pattern created inside of channels. The channels on the surface, represented in Figure 1a, are characterized by the following average measures: 0.348 mm of depth, 1.643 mm of length, 0.327 mm of width and 0.155 mm space between channels, while the holes had approximately 0.256 mm in diameter. Figure 1 shows that the laser has high-quality texturing because the channels and holes are well designed and do not show any kind of distortion. Moreover, all channels present equivalent size and spacing.

Due to the direct absorption of LASER energy that is influenced by energy density during LASER ablation, the removal of material from the specimen occurs [44]. This texturing is achieved by laser pulses that can be applied sequentially or in a burst. If applied in a burst, the material around the texture could remain hot for a longer period and the molten material could cover the texture [45,46]. Thus, the local heating of the Zr specimens plays an important role in the LASER texturing process. However, for this experiment the pulses were sequential.

No micro-cracks or heat-affected zones were observed, which suggests that texturing Zr surfaces in their non-sintered state is an effective way to produce different and complex textures. In addition, there are no defects that could compromise the mechanical properties of Zr. However, mechanical tests are needed to verify these assumptions. In Figure 2 the channels and holes filled with silver after hot-pressing are shown. Figure 2a depicts a top image, while in Figure 2b a profile image of the interior of the specimen is shown. In both cases, there are no discontinuities in the silver, suggesting that the impregnation occurred as expected and will allow the electric current flow. A representative Zr specimen was cut and polished to observe the pattern (Figure 2b). The cross-section image denotes the texture inside the channel, underscoring the effectiveness of the method to retain silver through the improvement of the contact surface area.

The top view image and the EDS are shown in Figure 3, and 2 zones are highlighted. These zones are Zr and the electric circuit (Ag), respectively, Z1 and Z2. The elements present on the EDS spectra are characteristic of the composition of each material corresponding to the analyzed zone (Zr − Z1 and Ag − Z2). The presence of carbon on both EDS spectra can be due to the hot-pressing technique, specifically because of the graphite dye. The EDS spectra of Z2 indicates the presence of silica, which can be explained by the polishing process.

### 2.2. Electric Current Application

The parameters were set to maintain a constant current equal to 6.5 mA, supplied to the specimens for 10 min. Given the data acquisition capability of the Gamry^®^ 600+, simultaneously with the current supply, it was possible to draw a representative curve of current as a function of time (Figure 4) and voltage as a function of time (Figure 5), which were effectively applied to the specimens. These data were acquired at a frequency of 100 Hz. Figure 4 and Figure 5 show only 3 s of the signal provided to the specimens; however, the signal, as already mentioned, remained constant. The maximum and minimum values were the same for all the specimens: for current, 6.57 mA and −6.54 mA, respectively, and for voltage, 3.29 × 10^−2^ V and −3.26 × 10^−2^ V, respectively.

### 2.3. Bacterial Adhesion

*S. aureus* was the bacterium selected to perform the tests due to its common presence in the oral cavity and because it is common in oral infections [47]. In addition, it is a Gram-positive bacterium. Gram-positive bacteria are usually reported to be more resistant to electrical stimulation than Gram-negative bacteria [48]. This enhanced resistance can be explained by the difference in the cell wall between Gram-positive and Gram-negative, namely peptidoglycan thickness and additional distinct cell wall components [48,49].

The effect of the application of the electric current was evaluated based on the bacterial adhesion to the surface of the specimens measured as concentration (CFU/mL). To estimate bacterial reduction, Equation (1) was used [50]. In Figure 6, it is possible to observe that the bacterium concentration is identical on the specimens Zr + Ag and Zr (nearly a 0.5 Log reduction).

The use of silver for medicinal purposes dates to a distant period of history, in which Ag and Ag salts were used to clean wounds and promote wound healing [26].

Ag is an inorganic material well known for its antimicrobial properties including against multiresistant microorganisms, satisfactory stability and non-toxic to humans when used in appropriate doses [51,52]. The Ag bactericidal mechanism is not fully understood under the conditions of the present study. Nevertheless, there are two main hypotheses related to endogenous and exogenous sources of reactive oxygen species to which cells can be exposed [53]. When metallic Ag is oxidized to Ag^+^, it can generate reactive oxygen species (ROS). Ag^+^ has been also reported to generate superoxide radical (O_2_^−^). These species, especially ROS, have a high preponderance in cell death by some mechanisms, including DNA damage culminating in extensive mutagenesis that leads to irreversible damage and, ultimately, the loss of viability. The induction of an apoptotic-like process and the inhibition of essential metabolic processes are other mechanisms of cell death by means of ROS [54,55]. It is also reported that silver ions cause the separation of paired DNA strands in bacteria and weaken the bond between protein and DNA, and ROS are generated as a consequence of this phenomenon [53]. ROS toxicity towards animal cells and bacteria is differentiated, mainly due to higher availability of resources and strategies for ROS mitigation of animal cells in comparison to the bacteria cell. Therefore, ROS typically exhibit lower toxicity towards animal cells [56,57]. Furthermore, the oxidation of metallic Ag into Ag^+^ is a slow process [58], which may be favorable to prevent metallic cytotoxicity, further supporting the use of Ag in a dental implant application.

Despite the insignificant differences in this study, the mechanisms reported are mainly responsible for the antibacterial activity of silver, and some of them could be related to this low decrease in bacterial adhesion to the specimens.

Regarding the specimens subjected to the application of alternating current, it was not possible to observe any CFU, which indicates that the application of this current can induce approximately a 7 Log reduction of adhered bacterium in comparison to Zr + Ag. This underscores the sterilizing activity of the specimen when current is applied.

In the oral cavity, teeth and dental implants are highly prone to bacterial colonization. The bacteria that adhere to the teeth and implant surface can form complex structures named biofilms [7]. The first step of biofilm formation is the adhesion of the microbes onto the surface. After this initial attachment and depending on the environmental conditions, the bacteria begin to colonize and grow on the implant surface. Extracellular Polymeric Substances (EPS) are secreted and distributed between cells, interacting with each other and forming a matrix that encompasses microbial cells. This matrix is important to act as a protective barrier against external forces. Subsequently, the attached bacteria begin to multiply and form microcolonies, culminating in biofilm maturation [59,60].

Successful treatment of these biomaterial-associated infections, such as peri-implantitis, depends on the extent of bacterial eradication. However, the growth of bacteria in biofilms greatly enhances their resistance to several types of treatment, such as the use of antibiotics [34,35,61]. The extensive literature reports that either stimulation DC or AC, or the use of high energy electric pulse, also called pulsed electric field (PEF), hinders bacterial adhesion, thus preventing biofilm formation [34,35,61].

On an electrically conductive surface to which a current or potential is applied, the adhesion of bacterium is essentially due to three types forces: electrostatic, electrophoretic and electroosmotic [62]. Individual bacteria possess a negative surface charge [63]. Thus, opposite charges are present on the specimen surface and the bacteria. Hence, an electrostatic interaction will promote the adhesion of bacteria to the specimen surface. Generally, this effect is avoided by bacterial inactivation which results from electric current effects [64].

Fish et al. [65] established the limit of 10.5 mA as the hazardous current limit to the human body. Thus, the current values used should be acceptable for use with humans. Guffey and Asmussen [66] and Barranco et al. [67] found an antibacterial effect with continuous currents against *S. aureus*. Barranco [67] demonstrated the death of *S. aureus* cells using 40 mA DC delivered with platinum, gold, Ag or stainless steel electrodes. For AC, Beattie [68] reported that a biocidal effect of AC can be obtained when a very high potential (3000–4000 V) was applied to sterilized milk. Later, in 1965, Rosenberg [69], in experiments with 2 A of AC [70,71], reported the death of *E. coli*. The use of 10 to 200 mA of AC for 10 s denoted a reduction in *E. coli* according to Pareilleux et al. [72]. In general, larger values of AC are required when compared to DC to obtain similar results [61].

In the present study, the use of 6.5 mA of AC for 10 min demonstrated the ability to inhibit *S. aureus* adhesion onto the specimens, which is in line with reported trends [38,49,73,74]. It was not possible to find any CFU in the specimens treated with AC. Several mechanisms can have contributed to this inhibition, which can be divided into direct effects of the electrical current or indirect effects [65].

Regarding the direct effects, these may be due to the disruption of the bacterial cell membrane or the block proliferation of bacterial cells [49,64]. Regarding indirect effects, there is the variation of pH and temperature, Ag^+^ ion release and actuation and the formation of gas bubble radicals that can lead to the detachment of bacteria, resulting from electrolysis [75,76]. As only bacterial adhesion was assessed, detachment from the gas bubbles or electrorepulsive forces may have been a determining factor.

Although the results obtained for AC are promising, further experiments should be carried out with other types of bacteria and electrodes. Furthermore, as the DC intensity depends on several factors such as polarity, electrode resistance, temperature, pH and electrolyte composition, among others, these parameters should be monitored [49]. Finally, testing the currents used in human cells (in vitro), to verify their behavior, should also be considered in future studies.

### 2.4. Metabolic Activity and Cell Morphology Analysis

Zr + Ag has emerged with the purpose of effectively inhibiting bacteria, while at the same time being non-toxicity to human cell [77]. Despite several studies already reporting the cytocompatibility of Zr + Ag and considering the potential application of these materials for bone tissue approaches [78,79], in the present study, Zr + Ag specimens, as well as Zr and Ag specimens, were cultured with hFOB. The aim was to understand the cell behavior in the different materials and access the Zr + Ag non-toxicity. On this basis, MTS results elucidate about the cells’ metabolic activity, which consequently can be transduced in cell viability and the results are displayed in Figure 7.

The MTS attested the viable metabolic activity of hFOB cells after 72 h in contact with all the tested materials, showing that all the produced specimens did not exhibit cytotoxicity levels. It is noteworthy that the Zr + Ag specimen presented higher metabolic activity. Cell adhesion and spreading were visualized by scanning electron microscopy (SEM), as depicted in Figure 8. It is possible to observe the unseeded (Figure 8a) and seeded specimen (Figure 8b,c). After the seeding of hFOB, it was possible to observe that cells seem to be adhered to the surface, especially in the interface Zr + Ag.

## 3. Materials and Methods

### 3.1. Zirconia Specimens Production

A commercial Yttria-stabilized Zr (TZ-3YB-E) powder with a uniform dispersion of 3 mol% Yttria (Tosoh Corporation, Tokyo, Japan) with high purity (99%) and theoretical density of 6.05 g/cm^3^ was used in this study to produce Zr specimens. This powder is constituted by spherical granules with an average size of 60 μm and contains small crystallites that are about 40 nm in diameter. The chemical composition and the scanning electron microscopy (SEM) image of Zr powder are presented in Table 1 and Figure 9.

The Zr specimens were produced by a powder metallurgy (PM) process, namely press and sintering, as schematically represented in Figure 10. The compaction of 3Y-TZP (TZ-3YB-E Tosoh, Tokyo, Japan) powder was performed on a cylindrical steel mold with an internal diameter of 10 mm. In the first step, the powder was introduced into the mold and a pressure of 200 MPa was applied for 30 s. Subsequently, the pressure was released evenly and compacted specimens (also designated by green bodies) with 10 mm of diameter and 3 mm of thickness were obtained.

### 3.2. Silver-Based Electric Circuit Production

#### 3.2.1. Zirconia Specimens Laser Texturing

Following the production of green compacts Zr, channels and holes were designed and produced on these specimens using an Nd:YAG laser (OEM Plus, SISMA, San Diego, CA, USA) to create an electric circuit structure. The characteristics of the laser used are listed in Table 2.

The design of the electric circuit structure (holes and channels) was defined in a computer-aided design system and then it was engraved on the green compacts Zr, according to the specific laser parameters mentioned in Table 3. The laser texturing process was performed in normal air under atmospheric pressure. A jet of air braid was used to remove the resultant debris produced during the process that could affect the surface texturing. In Figure 11 is provided a schematic representation of laser texturing regarding electric circuit structure production. Table 3 presents the laser parameters used to produce the different structures, channels, holes and patterns.

After laser texturing, Zr specimens were sintered using a high-temperature furnace (Zirkonofen 700, Zirkonzahn, South Tyrol, Italy) for 2 h at 1500 °C, in air, with a heating and cooling rate of 8.3 °C/min. The obtained specimens had the following average dimensions: 8 mm in diameter and 2 mm in thickness. The temperature was chosen based on the sintering temperatures (T = 1350–1550 °C) required by the supplier (Tosoh Corporation, Japan) for conventional sintering processes.

#### 3.2.2. Silver Impregnation

Ag powder (≥99 wt% pure) with an average grain size of 82 µm, from Metalor Technologies, Attleboro, MA, USA, was used to impregnate channels and holes to provide the specimens conductivity properties. This process was carried out by a powder metallurgy technology, namely, hot-pressing in vacuum (6 to 10 Pa) conditions and using a high-frequency induction furnace, as schematically represented in Figure 12. The mold was then placed inside the chamber, where the specimen was compressed at the required pressure (0.25 MPa) and heated up to around 1000 °C, with a heating rate of 77.8 °C/min. The specimen was maintained above 1000 °C and under the respective pressure for 2 min. Afterwards, the specimens were cooled inside the mold, in a vacuum, until reaching room temperature.

After silver impregnation on channels and holes, the specimens were polished to remove some excess silver, and then, using a conventional voltmeter, the electric current flow was verified. To facilitate electrical current application on in vitro tests, a silver wire was welded on each channel of the back face of the specimens. Zr as sintered and Zr with silver not stimulated with current were also tested for comparison purposes.

#### 3.2.3. Electric Configuration and Inuts

To perform in vitro tests, the electric configuration present in Figure 5 was used. A Gamry^®^ 600+ device was used to provide the electric current.

According to the work reported by Mohn et al. [37], current values of 2, 5, 7.5 and 10 mA were used for 10 min, and in the work of Sahrmann [33], a current value of 10 mA was used for 15 min. Given the characteristics of the available device and based on these references, an alternating current of 6.5 mA was applied for 10 min. The current value was fixed whereas the voltage was permitted to vary. The Virtual Front Panel application was used to generate the current signal.

### 3.3. Specimen Characterization

Specimen morphological characterization was analyzed by SEM (JEOL JSM-6010LV). Energy-dispersive X-Ray spectroscopy analysis (EDS) using FEG SEM, FEINova 200, Mahwah, NJ, USA equipment was used in some specific locations for chemical composition analysis of the specimens.

### 3.4. Specimen Sterilization

Prior to in vitro tests, the specimens were immersed in absolute ethanol for 2 min. Then, they were pulverized with distilled water and immersed in an ultrasonic bath (250 W and 50 KHz, J.P. Selecta, Barcelona, Spain) for 30 min. Subsequently, the specimens were rinsed in distilled water three times. Afterwards, the specimens were immersed in ultra-pure water for 20 min, transferred to a Petri dish (without lid) followed by exposure to UV-C for 20 min inside a biological safety level 2 air flow chamber. Subsequently, the specimens were transferred under sterile conditions to a 2mL Eppendorf containing absolute alcohol (1.5 mL) for 48 h. Finally, the specimens were left to dry for 24 h at room temperature inside an air flow chamber.

### 3.5. Staphylococcus aureus Adhesion Protocol

A single colony of *S. aureus* was collected from a solid medium (tryptic soy agar (TSA)) Petri dish incubated previously, and aseptically immersed in tryptic soy broth (TSB). It was incubated overnight at 37 °C and 120 rpm of shaking speed. Inoculum concentration was adjusted to approximately 1.25 × 10^7^ CFU/mL (spectrophotometer optical density (OD) at 600 nm of 0.123) in TSB. The specimens were inoculated with 200 µL on 24-well plates. Then the bacterium was incubated for 24 h at 37 °C and 120 rpm shaking speed, as our objective was to actuate at early stages of biofilm development. Afterwards, the electric current was applied. Three specimens per condition were tested: as sintered Zr specimens, zirconia with silver (Zr + Ag), and zirconia specimens with silver submitted to alternating current (Zr + Ag(AC)). In addition, a control group comprising bacteria incubated in a polystyrene plate (Ct).

Bacterium adhesion on the specimens surface was estimated through concentration determination. First, the medium in the wells with specimens was collected and discarded. Each specimen was carefully transferred into a sterile tube containing 6 mL of phosphate buffered saline (PBS) and ultrasonicated for 10 min 250 W and 50 KHz (J.P. Selecta, Spain), then vortexed at maximum speed for 1 min to detach bacteria adhered to the specimen’s surface. Then, serial dilutions were performed in PBS and the bacterium concentration was assessed through plating in Petri dishes containing TSA, and incubated at 37 °C for 24 h.

The adhesion was estimated according to Equation (1):(1)$$Log\;reduction\left(CFU/mL\right)=Log\left[control\left(CFU/mL\right)\right]-Log[exposed\;(CFU/mL)]$$

### 3.6. Cytotoxicity Evaluation Protocol

#### 3.6.1. Cell Culture

The human fetal osteoblasts cell line (hFOB 1.19; ATCC^®^, Manassas, VA, USA) was cultured in a 1:1 mixture of Ham’s F12 Medium Dulbecco’s Modified Eagle’s Medium without phenol red (DMEM:F12; PanBiotech^®^, Bavaria, Germany) and supplemented with 2.5 mM/L glutamine PanBiotech^®^, Bavaria, Germany), 0.3 mg/mL G418 (PanBiotech^®^, Bavaria, Germany) and 10% (*v*/*v*) of fetal bovine serum (FBS; Sigma-Aldrich, Darmstadt, Germany). The hFOBs were incubated at 37 °C in a humidified 5% CO_2_ atmosphere until a 90% confluency was reached. The passages used in this study were in the range of P6–P12, according to the manufacture’s indication.

The top surface of each sterilized specimens were seeded with a droplet of 50 μL comprising cell suspension of 30,000 hFOBs. After 3 h of cell adhesion, 300 µL of culture medium was added to each sample and kept in the incubator. After 3 days (72 h), the seeded specimens and the culture medium were collected to determine metabolic activity and cell morphology.

#### 3.6.2. Metabolic Activity

The metabolic activity of hFOB was measured by the 3-(4,5-dimethylthiazol-2-yl)-5-(3-carboxymethoxyphenyl)-2-(4-sulfophenyl)-2H-tetrazolium (MTS) cell proliferation colorimetric assay kit (BioVision, Piscataway, NJ, USA). Each specimen was transferred to a new 48-well plate containing 300 µL of culture medium, and 30 µL of reagent was added to each well and placed for 4 h in the incubator. Then, the volume of each well was transferred in duplicate in a 96-well plate and the absorbance was read at 490 nm in a microplate reader (Biotek, Winooski, VT, USA). It is noteworthy that a blank was prepared by incubating the dye reagent with only culture medium to subtract the absorbance of each sample.

#### 3.6.3. Cell Morphology Analysis

SEM (JEOL JSM-6010LV) was used to evaluate the cells adhesion and morphology seeded on the specimens. The specimens were collected and fixed with a 2.5% (*v*/*v*) glutaraldehyde (Sigma, USA) solution in PBS. Then they were washed with ultra-pure water and dehydrated in increased concentrations of ethanol from 30% (*v*/*v*) to 100% (*v*/*v*) and air-dried overnight. Prior to SEM observation, the specimens were coated with gold (gold sputtering equipment: Quorum/Polaron model E 6700; acceleration voltage of 5.00 kV).

### 3.7. Statistical Analysis

Data were analyzed using the GraphPad^®^ Prism version 6 (GraphPad^®^ Software, San Diego, CA, USA). All data comprises the mean ± standard error of the mean. Shapiro–Wilk normality and KS normality tests were used to determine the normality distribution of all data. Statistical analysis encompasses one-way ANOVA, with Kruskal–Wallis with Dunn’s post hoc test.

## 4. Conclusions

Zirconia specimens with electric circuit were successfully produced through laser texturing followed by silver impregnation and sintering, and used for in-vitro tests with *S. aureus* bacterium. Within the limitations of this study, some conclusions can be drawn:The manufacturing process proved to be effective in the creation of the desired geometries for channels, holes, and patterns. Thus, laser texturing did not affect the integrity of the specimens, but further tests are needed to verify their mechanical properties. Furthermore, laser technology proved to be an effective method to produce structures on zirconia surfaces under non-sintered states.The results showed that silver impregnation allowed the conduction of electric current and did not show visible discontinuities. The specimen’s in vitro cytocompatibility were screened using hFOB cells showing that the Zr + Ag specimen was non-cytotoxic and was able to host cells for the time of culture.The use of silver, under the conditions of the present study, has shown a non-significant reduction in bacterial adhesion (≈0.5 log reduction).The application of an alternating electric current of 6.5 mA showed a potential inhibition of adhesion of the specimens for the present study (≈7 log reduction).It can be concluded that the application of electric current can be an alternative and promising strategy to avoid and combat adhered bacteria that can evolve to biofilm, specifically in dental implants.

## Figures and Tables

**Figure 1 ijms-25-05719-f001:**
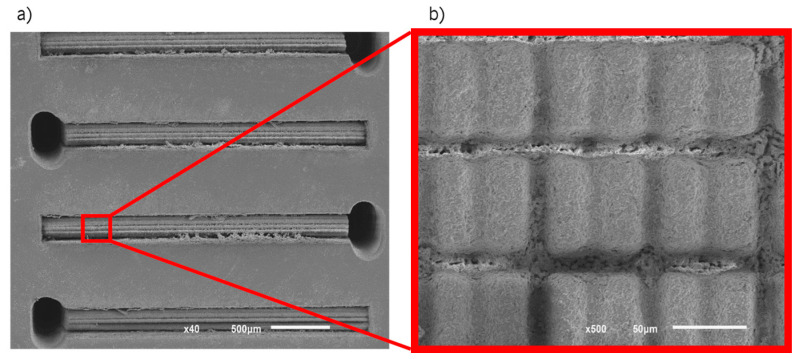
SEM micrographs obtained by laser texturing of the (**a**) top surface (**b**) pattern on channels bottom.

**Figure 2 ijms-25-05719-f002:**
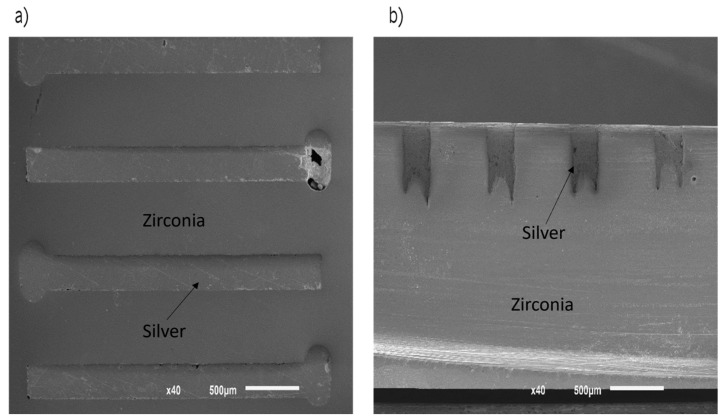
SEM micrographs obtained by silver impregnation (**a**) top surface (**b**) profile of the interior of the sample.

**Figure 3 ijms-25-05719-f003:**
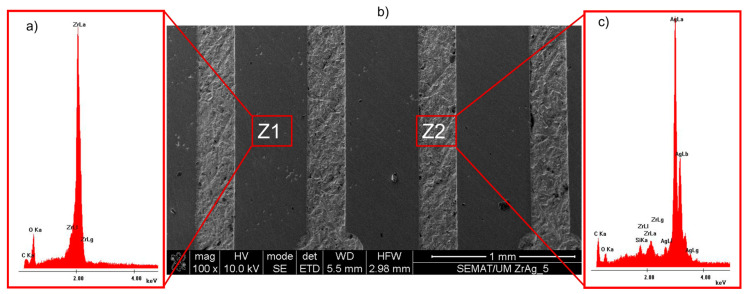
Top view SEM image (**b**) and EDS spectra of zone 1 (**a**); and zone 2 (**c**).

**Figure 4 ijms-25-05719-f004:**
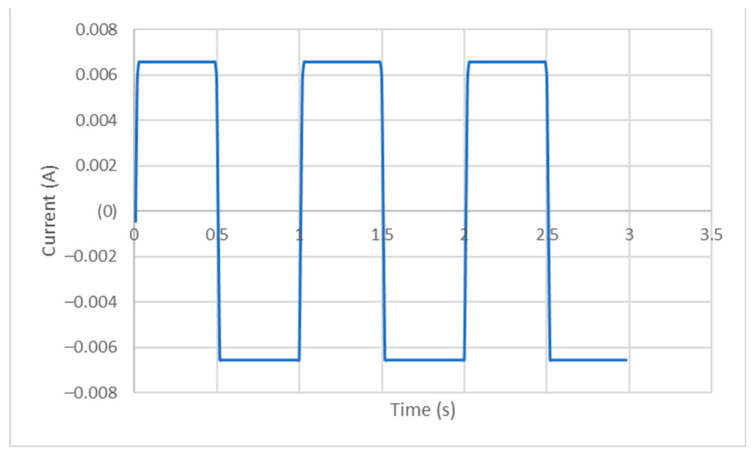
A representative curve of current variation as function of time of application.

**Figure 5 ijms-25-05719-f005:**
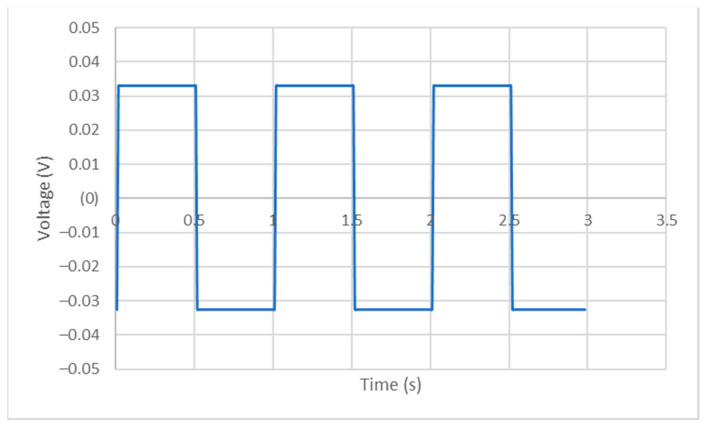
A representative curve of voltage variation as function of time of application.

**Figure 6 ijms-25-05719-f006:**
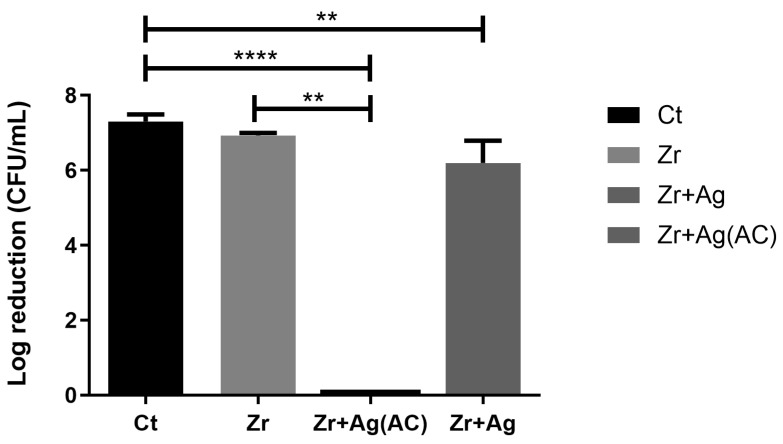
Bacterial adhesion (CFU/mL) on the tested surfaces after 24 h in contact with specimen’s surface and then treated with electrical current stimulus; ** corresponds to *p* < 0.01, and **** is *p* < 0.0001, (Zr: zirconia; Zr + Ag: zirconia with silver, Zr + Ag(AC): zirconia with silver submitted to alternating current and Ct-control on polystyrene plate).

**Figure 7 ijms-25-05719-f007:**
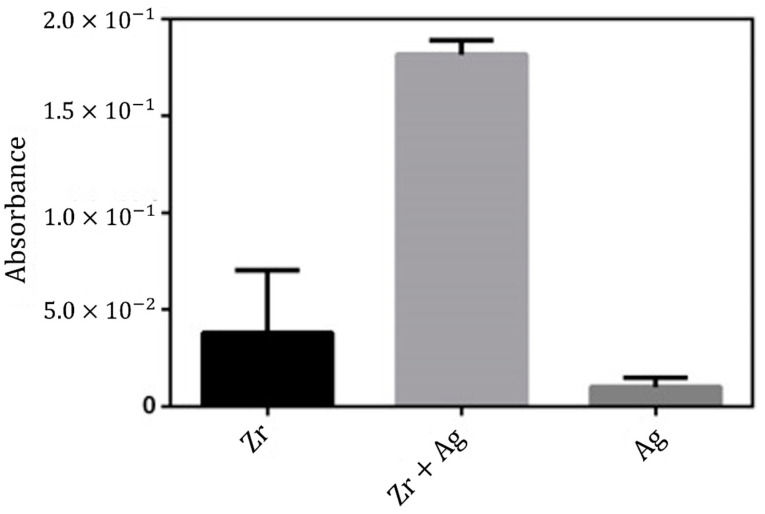
The hFOB metabolic activity after being cultured for 72 h in the Zr, Zr + Ag and Ag specimens.

**Figure 8 ijms-25-05719-f008:**
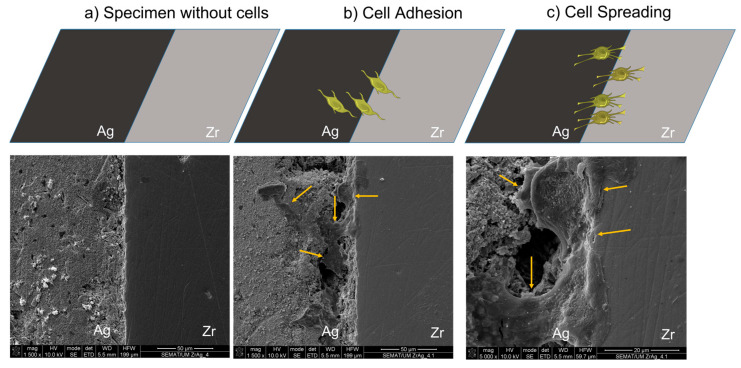
SEM analysis. SEM image of (**a**) unseeded specimen Zr + Ag; (**b**) seeded Zr + Ag; and (**c**) cell detail on seeded Zr + Ag. The inset yellow arrows highlight the hFOB elongation.

**Figure 9 ijms-25-05719-f009:**
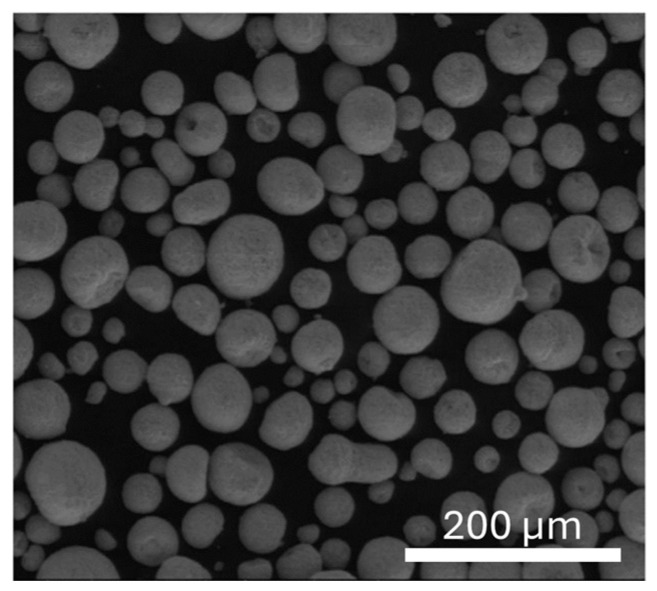
SEM image of TZ-3YB-E powder.

**Figure 10 ijms-25-05719-f010:**
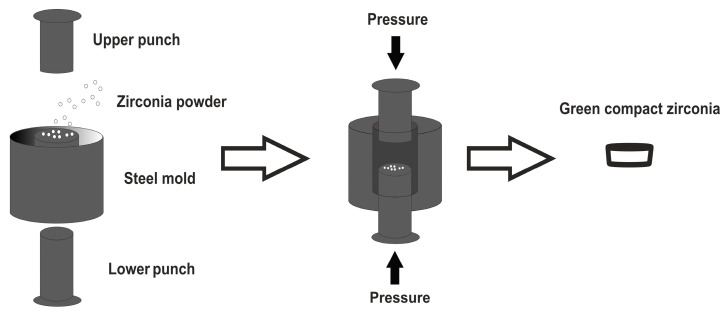
Schematic illustration of zirconia specimens’ production.

**Figure 11 ijms-25-05719-f011:**
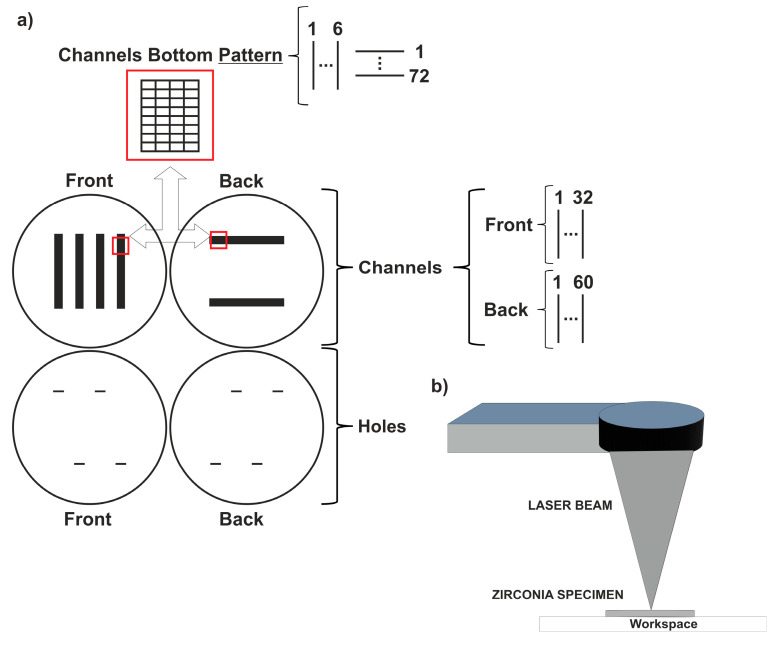
Schematic representation of the experimental set-up of zirconia specimens laser texturing: (**a**) 2D drawing parameters and (**b**) laser texturing.

**Figure 12 ijms-25-05719-f012:**
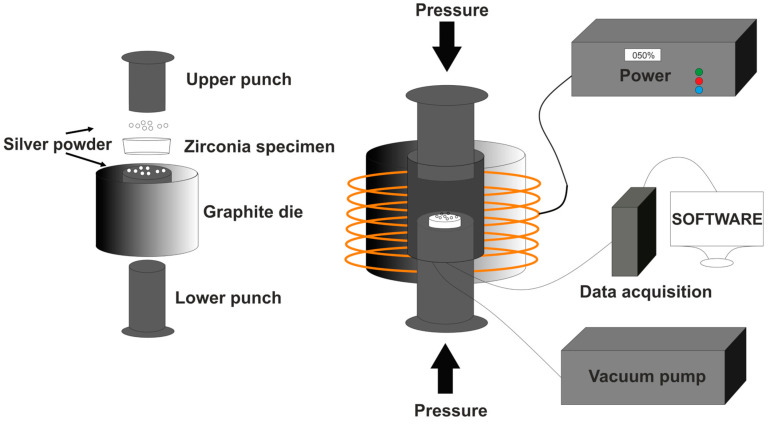
Schematic representation of hot-pressing technique.

**Table 1 ijms-25-05719-t001:** Chemical composition of TZ-3YB-” powder according to Tosoh Corporation.

Elements	TZ-3YB-E
Al_2_O_3_ (wt%)	0.1–0.4
Fe_2_O_3_ (wt%)	≤0.01
Na_2_O (wt%)	≤0.04
SiO_2_ (wt%)	≤0.02
Y_2_O_3_ (wt%)	5.2 ± 0.5
HfO_2_ (wt%)	<5.0
ZrO_2_ + HfO_2_ + Y_2_O_3_ (wt%)	(>99.8)

**Table 2 ijms-25-05719-t002:** Nd:YAG laser (OEM Plus, SISMA, Italy) characteristics.

Output power	6 W
Spot size	3 μm
Pulse width	≈35 ns
Nominal focal length	160 mm
Fundamental wavelength	1.064 μm
Maximum pulse energy	0.3 mJ/pulse

**Table 3 ijms-25-05719-t003:** Laser parameters used to produce the different structures.

Structure		Power (%)	Number of Passages	Scan Speed (mm/s)	Wobbel Amplitude
Channels	Front	25	10	128	0
	Back		8		
Holes		25	8	From 20 to 2 every 2	From 0.300 to 0.030 every 0.030
Pattern	Vertical lines	25	4	128	0
	Horizontal lines		8		

## Data Availability

Data is contained within the article.

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
