# Peer review of "Zirconia Dental Implants Surface Electric Stimulation Impact on Staphylococcus aureus"

_ijms, 2024, doi:10.3390/ijms25115719_

Round 1
Reviewer 1 Report
Comments and Suggestions for Authors
After reading the article, the main postulate of the authors that their material does not emit Ag ions seems doubtful. If the material does not emit ions, then how can the bactericidal nature of the material be determined.
Line 200 The Ag bactericidal mechanism is not fully understood [ 61] This statement is erroneous. In this article there is nothing at all about silver, it is generally about something else (H. Wag, Wireless Electrostimulation to Eradicate Bacterial Biofilms, Diss. - ALL. (2019). It is now known that silver ions cause the separation of paired DNA strands in bacteria and the weakening of the bond between protein and DNA. Calorimetric analyses have confirmed that DNA and silver ions interact directly. In this case, the faster protein dynamics caused by silver can be explained. When a protein is bound to DNA, it moves slowly in bacteria along with DNA, which is a huge molecule. On the contrary, when treated with silver, proteins fall away from DNA, moving by themselves and, therefore, moving faster. And the ROS mentioned by the authors are a consequence of these processes, but not the cause.
Line 185 - S. aureus was the selected bacterium to perform the tests due to its common presence in the oral cavity and is a common responsible for oral infections [47] This opportunist can cause infections of the oral cavity, but other species responsible for formation of biofilms used in the study of peri-implantitis. Moreover, model experiments are performed under anaerobic conditions, as it happens in vivo. It is known that the microbiological picture in peri–implantitis is similar to that in periodontitis: Prevotella intermedia is detected in 100%; Porphyromonas gingivalis – in 89%; Actinobacillus actinomycetemcomitans – in 85%; Bacteroides forsythus - 55%; Treponema denticola – in 41%. And this is not done with a certain logic. The formation and functioning of their community and biofilms is very different from that of Staphylococcus aureus. I believe that the species was chosen incorrectly for this purpose, as well as the cultivation conditions. Line 419 - is too short a time for the formation of a stable biofilm, shaking can disrupt the film formation process. It is also a questionable choice to consider CFU for bacteria in biofilm. With a normally formed biofilm, bacteria are immobilized in it, as well as mucous capsules of exopolysaccharides and proteins cause bacteria to stick together in microcolonies. In such experiments, it is necessary to use fluorescence microscopy with counting cells inside the matrix if you want to roughly count the number of bacteria. If you used microscopy of bacteria in your work, you would see the voiced nuances.
Line 193 Erro! A origem da referência não foi encontrada What is it?
Line 238 Bacteria possess a negative surface charge [63]. And biofilms have nothing to do with it, biofilm is not a bacterial wall capsule, the following logical conclusions are questionable
Lines 263-266 Regarding indirect effects, there is the variation of pH and temperature, and production of toxic substances, e.g. Ag+ ions, and the formation of gas bubbles radicals that can lead to detachment of bacteria, resulting from electrolysis [73,74] And again. According to your statement, the material does not emit ions. Then how is it bactericidal? It seems to me that in vivo this material can release ions under the influence of organic acids of the bacterial film. Then it will be bactericidal.
Lines 286-292 This is an assessment of the non-toxicity of the material. But the work is about stimulation. Electrical stimulation can lead to the death of gum tissue around the implant. As far as I understand, you have not checked this, and without this, it is extremely doubtful to assert that this method is suitable for the treatment of peri-implantitis and not damage to surrounding tissues. It is also necessary to carry out a microscopic assessment of living cells using appropriate fluorescent probes. And again, it is known that silver ions are cytotoxic for animal cells as well. Without fluorescence microscopy, this looks inconclusive.
Fig7 The Absorvanse (not English?) of Zr and Ar almost completely falls into error, why is that?
Table 2 cell offset
Line 413 3.4 to 3.5
Line 439 Citotoxicity to Cytotoxicity
Line 456 Cell morphology analysis
To fix the cells, you need to use an arsenic buffer. Post-fixation of fats and membranes is carried out by osmium tetracoide. And after dehydration in a series of alcohols, it is necessary to dry not in air, but either in a freeze dryer with ethanol substitution for tretbutyl, or generally carry out the entire procedure of fixation and dehydration in glutar, osmium vapors and dehydration in propylene oxide vapors. The figure with electron microscopy of animal cells is extremely uninformative due to the low resolution, it is difficult to assess what the authors claim. And it is impossible to evaluate the morphology of animal cells, especially in the context of viability, without post-fixation with osmium and normal dehydration.
Line 496-498 For such a statement, it is necessary to study with the help of at least a set of living dead bacterial cells after exposure. The adhesion may have changed for other reasons, although the bacteria remain alive.
Line 499 Only for Staphylococcus aureus, which is not the main type in the problem of peri-implantitis. And it is unknown how this effect will affect the tissues surrounding the implant.
It is not in my competence to evaluate the physical part of the article, but the biological part, conclusions and selection of methods look questionable. I believe that it is impossible to publish the material in this form. The authors need to study in more depth the process of peri-implantitis formation, the specificity of bacteria and the mechanism of Ag action on living cells. Also, to supplement the methods for determining cytotoxicity with fluorescent probes, as a necessary standard for their approval. It is necessary to show the biofilm and the adhesion disorder using SAM or TEM, as well as to show the death of bacterial cells using fluorescent probes. And also to determine the Ag ions and their possible diffusion from the material, or not to assert its impossibility.
Author Response
Response to Reviewer 1 Comments
|
||
1. Summary |
|
|
We appreciate the opportunity to revise our manuscript based on the Reviewers' feedback. Thank you for the time and effort put into providing thoughtful comments and suggestions which greatly improved the impact of our manuscript. Below, we address each of the Reviewers' concerns and have made corresponding revisions, which are highlighted in the re-submitted files.
|
||
2. Point-by-point response to Comments and Suggestions for Authors |
||
Comment 1 After reading the article, the main postulate of the authors that their material does not emit Ag ions seems doubtful. If the material does not emit ions, then how can the bactericidal nature of the material be determined.
|
||
Response 1 Thank you for your thoughtful review of our article. We appreciate your attention to detail and your insightful comment regarding a main postulate about the emission of Ag ions. We recognize your concern about how the bactericidal nature of the material can be determined if it does not emit ions. It's important to note that our understanding is that the release of ions occurs once the material is in contact with an aqueous medium and is subjected to electric current. We recognize that in this specific study, we did not quantify the concentration of ions released into the medium, but we are aware that this ion release occurs and contributes to bactericidal phenomenon alongside other mechanisms.
|
||
Comment 2 Line 200 The Ag bactericidal mechanism is not fully understood [ 61] This statement is erroneous. In this article there is nothing at all about silver, it is generally about something else (H. Wag, Wireless Electrostimulation to Eradicate Bacterial Biofilms, Diss. - ALL. (2019).
It is now known that silver ions cause the separation of paired DNA strands in bacteria and the weakening of the bond between protein and DNA. Calorimetric analyses have confirmed that DNA and silver ions interact directly. In this case, the faster protein dynamics caused by silver can be explained. When a protein is bound to DNA, it moves slowly in bacteria along with DNA, which is a huge molecule. On the contrary, when treated with silver, proteins fall away from DNA, moving by themselves and, therefore, moving faster. And the ROS mentioned by the authors are a consequence of these processes, but not the cause. |
||
Response 2 Thank you for the feedback on our article. Indeed, the reference provided does not align with the content of our article. It was an error during referencing. We have corrected it. We deeply acknowledge the thorough review.
LINE 200:
Thank you for bringing this to our attention. After considering the comment, we've found that there is no consensus in the literature regarding this topic, due to the enormous challenges to precisely observe what is in fact occurring inside the cells. There are two main hypothesis “Cells are exposed to both endogenous and exogenous sources of reactive oxygen species (ROS).” (Rowe LA, Degtyareva N, Doetsch PW. DNA damage-induced reactive oxygen species (ROS) stress response in Saccharomyces cerevisiae. Free Radic Biol Med. 2008 Oct 15;45(8):1167-77. doi: 10.1016/j.freeradbiomed.2008.07.018. Epub 2008 Jul 30. PMID: 18708137; PMCID: PMC2643028.) 1. The hypothesis that reviewer mentioned, which suggests that silver ions cause the separation of paired DNA strands in bacteria and weaken the bond between protein and DNA and because of that ROS are generated. This hypothesis is supported by references such as: Rowe LA, Degtyareva N, Doetsch PW. DNA damage-induced reactive oxygen species (ROS) stress response in Saccharomyces cerevisiae. Free Radic Biol Med. 2008 Oct 15;45(8):1167-77. doi: 10.1016/j.freeradbiomed.2008.07.018. Epub 2008 Jul 30. PMID: 18708137; PMCID: PMC2643028. 2. However, according to other studies, the damage caused to DNA is partly due to reactive oxygen species (ROS). To reinforce this information, we have added additional references, that are also addressed below: Kessler, A.; Hedberg, J.; Blomberg, E.; Odnevall, I. Reactive Oxygen Species Formed by Metal and Metal Oxide Nanoparticles in Physiological Media—A Review of Reactions of Importance to Nanotoxicity and Proposal for Categorization. Nanomaterials 2022, 12, 1922. https://doi.org/10.3390/nano12111922 Brinkman CL, Schmidt-Malan SM, Karau MJ, Greenwood-Quaintance K, Hassett DJ, Mandrekar JN, et al. (2016) Exposure of Bacterial Biofilms to Electrical Current Leads to Cell Death Mediated in Part by Reactive Oxygen Species. PLoS ONE 11(12): e0168595. https://doi.org/10.1371/journal.pone.0168595 Stewart PS, Wattanakaroon W, Goodrum L, Fortun SM, McLeod BR. Electrolytic generation of oxygen partially explains electrical enhancement of tobramycin efficacy against Pseudomonas aeruginosa biofilm. Antimicrob Agents Chemother. 1999 Feb;43(2):292-6. doi: 10.1128/AAC.43.2.292. PMID: 9925521; PMCID: PMC89066. We acknowledge that the way it was originally written may have been misleading. This section was rewritten and new references were added to the manuscript to improve the quality and clearity. LINE 201:
|
||
Comment 3 Line 185 S. aureus was the selected bacterium to perform the tests due to its common presence in the oral cavity and is a common responsible for oral infections [47] This opportunist can cause infections of the oral cavity, but other species responsible for formation of biofilms used in the study of peri-implantitis. Moreover, model experiments are performed under anaerobic conditions, as it happens in vivo. It is known that the microbiological picture in peri–implantitis is similar to that in periodontitis: Prevotella intermedia is detected in 100%; Porphyromonas gingivalis – in 89%; Actinobacillus actinomycetemcomitans – in 85%; Bacteroides forsythus - 55%; Treponema denticola – in 41%. And this is not done with a certain logic. The formation and functioning of their community and biofilms is very different from that of Staphylococcus aureus. I believe that the species was chosen incorrectly for this purpose, as well as the cultivation conditions.
|
||
Response 3 We completely agree with the Reviewer, in fact that is one of the main envisaged objectives. We started our analysis with a facultative aerobe, simply due to operational conditions using standard culture conditions. S. aureus allowed us to prove the feasibility and adequacy of the developed methods under aerobic conditions. Thus, we can extrapolate the same methods for anaerobic condition which are more complex in operational terms, and more time consuming. The next step will be to perform the same methods under anaerobic conditions using some bacteria mentioned by the Reviewer, inside an anaerobic chamber. The rich culture media will also be gradually replaced by a more representative medium of oral cavity environment, namely artificial saliva. Therefore, all relevant bacteria will be tested to fully understand the effect of this electric current treatment. In addition, S. aureus is a well-known and extensively studied bacterial species, particularly due to its status as a multi-resistant bacterium. Furthermore, this bacterium was used in similar contexts, making possible its comparison with the available literature.
Wang H, Ren D. Controlling Streptococcus mutans and Staphylococcus aureus biofilms with direct current and chlorhexidine. AMB Express. 2017 Nov 15;7(1):204. doi: 10.1186/s13568-017-0505-z. PMID: 29143221; PMCID: PMC5688048.
Minkiewicz-Zochniak A, Strom K, Jarzynka S, Iwańczyk B, Koryszewska-Bagińska A, Olędzka G. Effect of Low Amperage Electric Current on Staphylococcus Aureus-Strategy for Combating Bacterial Biofilms Formation on Dental Implants in Cystic Fibrosis Patients, In Vitro Study. Materials (Basel). 2021 Oct 15;14(20):6117. doi: 10.3390/ma14206117. PMID: 34683710; PMCID: PMC8537792.
Zituni, D., Schütt-Gerowitt, H., Kopp, M. et al. The growth of Staphylococcus aureus and Escherichia coli in low-direct current electric fields. Int J Oral Sci 6, 7–14 (2014). https://doi.org/10.1038/ijos.2013.64
|
||
Comment 4 Line 419 Is too short a time for the formation of a stable biofilm, shaking can disrupt the film formation process. It is also a questionable choice to consider CFU for bacteria in biofilm. With a normally formed biofilm, bacteria are immobilized in it, as well as mucous capsules of exopolysaccharides and proteins cause bacteria to stick together in microcolonies. In such experiments, it is necessary to use fluorescence microscopy with counting cells inside the matrix if you want to roughly count the number of bacteria. If you used microscopy of bacteria in your work, you would see the voiced nuances. |
||
Response 4 We would like to acknowledge the Reviewer insightful comment regarding the lack of staibility, maturation of the biofilm. This was the intended purpose, since our objective was to actuate at early stages of biofilm development, where interventions may be most effective in preventing further growth. Nevertheless, we failed to clearly state this in the manuscript, thus we added the following sentence: LINE 425: “Then, the bacterium was incubated for 24 h at 37 °C and 120 rpm shaking speed, since the objective was to actuate at early stages of biofilm development.” This aids to respond to the second observation performed by the Reviewer. The biofilm is in its early stages, so it is not difficult to adequately homogenize the bacterium. Thus, the CFU concentration is not hindered in any way. |
||
Comment 5 Line 193 Erro! A origem da referência não foi encontrada What is it?
|
||
Response 5 Thank you for your careful review. This consists in an error message by the referencing software that we missed to revise. It states that origin of the reference could not be found. However, we have revised the entire manuscript to avoid such messages. We apologize for any confusion this may have caused and appreciate your diligence in identifying this issue. |
||
Comment 6 Line 238 Bacteria possess a negative surface charge [63]. And biofilms have nothing to do with it, biofilm is not a bacterial wall capsule, the following logical conclusions are questionable.
|
||
Response 6 Thank you for your comment. As stated in Response 4, the biofilm is intended to be at its early stages of development, thus we consider that the negative surface charge will not be a negligible factor. LINE 425: “Then, the bacterium was incubated for 24 h at 37 °C and 120 rpm shaking speed, since the objective was to actuate at early stages of biofilm development.”
We have revised the statement for clarity to address your concern. The way this writing can be misleading and I have reformulated it to be clear. This is a behavior prior to the already formed biofilm, but rather the behavior of the bacteria to form the biofilm. Your insight is appreciated, and I've made the necessary adjustments to ensure clarity. Line 238: |
||
Comment 7 Lines 263-266 Regarding indirect effects, there is the variation of pH and temperature, and production of toxic substances, e.g. Ag+ ions, and the formation of gas bubbles radicals that can lead to detachment of bacteria, resulting from electrolysis [73,74]. And again. According to your statement, the material does not emit ions. Then how is it bactericidal? It seems to me that in vivo this material can release ions under the influence of organic acids of the bacterial film. Then it will be bactericidal. |
||
Response 7 Thank you for your comment. We recognize that ions release occurs and is partly responsible for the bacteria's death. However, the variation in pH, temperature, and other factors also contribute to this process and will be evaluated in future works, as well as measurements of ions release. This topic was revised and rewritten.
Line 268: “Regarding indirect effects, there is the variation of pH and temperature, and Ag+ ions release and actuation, and the formation of gas bubbles radicals that can lead to detachment of bacteria, resulting from electrolysis [75,76].”
|
||
Comment 8 Lines 286-292 This is an assessment of the non-toxicity of the material. But the work is about stimulation. Electrical stimulation can lead to the death of gum tissue around the implant. As far as I understand, you have not checked this, and without this, it is extremely doubtful to assert that this method is suitable for the treatment of peri-implantitis and not damage to surrounding tissues. It is also necessary to carry out a microscopic assessment of living cells using appropriate fluorescent probes. And again, it is known that silver ions are cytotoxic for animal cells as well. Without fluorescence microscopy, this looks inconclusive. |
||
Response 8 Thank you for your insightful comment. The cell assays were conducted primarily to evaluate the cytotoxicity of silver ions, as you rightly pointed out. We acknowledge the importance of considering the potential release and action of ions. However, we also understand your concern regarding the lack of assessment for electrical stimulation effects on gum tissue around the implant. This is indeed an important aspect that we plan to address in future studies. Regarding the suggestion for microscopic assessment of living cells using appropriate fluorescent probes, we appreciate the recommendation. We will consider incorporating fluorescence microscopy to provide a more comprehensive analysis in future research. Your feedback is valuable to us, and we will take it into consideration as we continue our investigation. Thank you for bringing these points to our attention. |
||
Comment 9 Fig7 The Absorvanse (not English?) of Zr and Ar almost completely falls into error, why is that? Table 2 cell offset Line 413 3.4 to 3.5 Line 439 Citotoxicity to Cytotoxicity
|
||
Response 9 Thank you for your appointment. All these items have been corrected to ensure accuracy. |
||
Comment 10 Line 456 Cell morphology analysis To fix the cells, you need to use an arsenic buffer. Post-fixation of fats and membranes is carried out by osmium tetracoide. And after dehydration in a series of alcohols, it is necessary to dry not in air, but either in a freeze dryer with ethanol substitution for tretbutyl, or generally carry out the entire procedure of fixation and dehydration in glutar, osmium vapors and dehydration in propylene oxide vapors. The figure with electron microscopy of animal cells is extremely uninformative due to the low resolution, it is difficult to assess what the authors claim. And it is impossible to evaluate the morphology of animal cells, especially in the context of viability, without post-fixation with osmium and normal dehydration
|
||
Response 10: We would like to thank the Reviewer for this comment. For the cell morphology analysis we follow a protocol that we consider to be an accepted standard. Unfortunately, we do not know about the Arsenic buffer method mentioned by the Reviewer, but we will consider it in future works. We would like to mention that we applied our method in our research group in recent publications:
Nevertheless, we revised the statement in an attempt to respond to the Reviewer concerns.
LINE 296: “After the seeding of hFOB, it was possible to observe that cells seems to be adhered to the surface, especially in the interface Zr+Ag.” |
||
Comment 11 Line 496-498 For such a statement, it is necessary to study with the help of at least a set of living dead bacterial cells after exposure. The adhesion may have changed for other reasons, although the bacteria remain alive. |
||
Response 11 Thank you for your valuable comment. We agree that further study, such as examining a set of living and dead bacterial cells after exposure, would provide a more comprehensive understanding. It's important to note that while we were able to detach the bacteria, this could indeed make it easier for a posterior elimination. We appreciate your insight and will be consider in future works. |
||
Comment 12 Line 499 Only for Staphylococcus aureus, which is not the main type in the problem of peri-implantitis. And it is unknown how this effect will affect the tissues surrounding the implant. |
||
Response 12 Thank you for your comment. We agree that Staphylococcus aureus may not be the main type of bacteria involved in peri-implantitis, and the effect on the surrounding tissues is unknown. The text has been rewritten to reflect this. LINE 500:
|
||
Comment 13 It is not in my competence to evaluate the physical part of the article, but the biological part, conclusions and selection of methods look questionable. I believe that it is impossible to publish the material in this form. The authors need to study in more depth the process of peri-implantitis formation, the specificity of bacteria and the mechanism of Ag action on living cells. Also, to supplement the methods for determining cytotoxicity with fluorescent probes, as a necessary standard for their approval. It is necessary to show the biofilm and the adhesion disorder using SAM or TEM, as well as to show the death of bacterial cells using fluorescent probes. And also to determine the Ag ions and their possible diffusion from the material, or not to assert its impossibility. |
||
Response 13 Thank you for your thorough evaluation and insightful comments on our article. We appreciate your expertise and understand the importance of a comprehensive understanding of peri-implantitis formation, bacterial specificity, and the mechanisms of silver action on living cells. We have taken your suggestions seriously and made significant revisions to address these concerns, and improve the manuscript quality. |

Reviewer 2 Report
Comments and Suggestions for Authors
It is a very detailed and significant study, but the proofreading of the text (Intro, M&M, Results, Discussion) is inadequate and there is no description of how this study can be used clinically as a treatment for peri-implantitis. In addition, there is a lack of discussion and explanation of the limitations of this study, which should be added.
Author Response
Response to Reviewer 2 Comments
|
||
1. Summary |
|
|
We are grateful for the opportunity to revise our manuscript based on the reviewers' feedback. Thank you for dedicating your time and effort to provide thoughtful comments and suggestions. Below, we address each of the reviewers' concerns and have made corresponding revisions, which are highlighted in the re-submitted files. |
||
3. Point-by-point response to Comments and Suggestions for Authors |
|
|
Comment 1 It is a very detailed and significant study, but the proofreading of the text (Intro, M&M, Results, Discussion) is inadequate and there is no description of how this study can be used clinically as a treatment for peri-implantitis. In addition, there is a lack of discussion and explanation of the limitations of this study, which should be added. |
||
Response 1 Thank you for your comment, which significantly contributes to improving the article. This study is primarily a proof of concept regarding the effect of electrical stimulation as a potential antibacterial agent in dentistry. Therefore, the level of current applied and the duration were chosen according to an extensive literature review (that results in a paper review under publication) and also limits that human body can tolerate. Currently, we are not dealing with direct clinical application, as further studies related to method of action and safety are needed. After the concept validation, studies in real-scale prototypes are required, followed by in vivo studies with animals, and only then considering clinical application. Thank you again for your valuable feedback.
|

Round 2
Reviewer 1 Report
Comments and Suggestions for Authors
The authors have seriously revised the article. And although many statements have not been verified by the methods I have proposed, but in this form the article can be published as is. It is likely that if the authors take into account many of the voiced points, the resulting material can be published as a new important article.
Reviewer 2 Report
Comments and Suggestions for Authors
The content is acceptable, with appropriate corrections.